# Influence of the Peripheral Nervous System on Murine Osteoporotic Fracture Healing and Fracture-Induced Hyperalgesia

**DOI:** 10.3390/ijms24010510

**Published:** 2022-12-28

**Authors:** Isabel Wank, Tanja Niedermair, Daniel Kronenberg, Richard Stange, Christoph Brochhausen, Andreas Hess, Susanne Grässel

**Affiliations:** 1Institute of Experimental and Clinical Pharmacology and Toxicology, Friedrich-Alexander-Universität Erlangen-Nürnberg, 91054 Erlangen, Germany; 2Institute of Pathology, University of Regensburg, 93053 Regensburg, Germany; 3Department of Regenerative Musculoskeletal Medicine, Institute of Musculoskeletal Medicine (IMM), University Hospital Münster, 48149 Münster, Germany; 4Centre for Medical Biotechnology (ZMB), Department of Orthopedic Surgery, Experimental Orthopedics, University of Regensburg, 93053 Regensburg, Germany

**Keywords:** osteoporosis, fracture healing, bone–brain nervous system interactions, fMRI sensory and sympathetic nervous system

## Abstract

Osteoporotic fractures are often linked to persisting chronic pain and poor healing outcomes. Substance P (SP), α-calcitonin gene-related peptide (α-CGRP) and sympathetic neurotransmitters are involved in bone remodeling after trauma and nociceptive processes, e.g., fracture-induced hyperalgesia. We aimed to link sensory and sympathetic signaling to fracture healing and fracture-induced hyperalgesia under osteoporotic conditions. Externally stabilized femoral fractures were set 28 days after OVX in wild type (WT), α-CGRP- deficient (α-CGRP −/−), SP-deficient (Tac1−/−) and sympathectomized (SYX) mice. Functional MRI (fMRI) was performed two days before and five and 21 days post fracture, followed by µCT and biomechanical tests. Sympathectomy affected structural bone properties in the fracture callus whereas loss of sensory neurotransmitters affected trabecular structures in contralateral, non-fractured bones. Biomechanical properties were mostly similar in all groups. Both nociceptive and resting-state (RS) fMRI revealed significant baseline differences in functional connectivity (FC) between WT and neurotransmitter-deficient mice. The fracture-induced hyperalgesia modulated central nociception and had robust impact on RS FC in all groups. The changes demonstrated in RS FC in fMRI might potentially be used as a bone traumata-induced biomarker regarding fracture healing under pathophysiological musculoskeletal conditions. The findings are of clinical importance and relevance as they advance our understanding of pain during osteoporotic fracture healing and provide a potential imaging biomarker for fracture-related hyperalgesia and its temporal development. Overall, this may help to reduce the development of chronic pain after fracture thereby improving the treatment of osteoporotic fractures.

## 1. Introduction

Treatments related to fracture repair and bone regeneration in compromised musculoskeletal conditions, such as osteoporosis, are in many ways a major issue in clinics. To optimize this situation, we need to understand the underlying principles of bone healing and regeneration. Numerous studies underline the crosstalk among cells of different tissues and origin and their importance for appropriate bone healing [1,2]. Especially, sensory and sympathetic nerve fibers are critically involved in timely and successful bone healing, as they frequently innervate the trabecular and cortical bone, periosteum and fracture callus. Sensory and sympathetic neurotransmitters contribute to the healing process by controlling bone metabolism, osteogenic differentiation, mineralization and remodeling processes [3,4,5,6], as well as modulating inflammation and nociception [7]. Furthermore, pro-inflammatory cytokines, neurotransmitters and growth factors released by macrophages invading the facture site are able to sensitize nerve endings, leading to intense acute pain and the development of hyperalgesia [8,9].

Chronic pain and subsequently poor quality of life are a common consequence of osteoporotic fractures [10]. Exploring the link between sympathetic and sensory neurotransmitter, fracture healing, and the development and resolution of hyperalgesia is crucial for optimizing the treatment of fractures and non-unions, especially under pathological conditions such as osteoporosis [1].

SP is released from capsaicin-sensitive primary afferent nerve endings and evokes a process termed ‘neurogenic inflammation’. This causes vasodilatation and plasma protein extravasation, regulates leukocyte chemotaxis and leads to sensitization of the primary neuron. It therefore plays a crucial role in transmitting nociceptive signals from the periphery to the CNS in acute and chronic pain [11,12,13,14]. Disruption of SP signaling impairs neurogenic inflammation and primary neuron sensitization [15,16]. Our previous studies showed that SP modulated bone resorption and bone formation processes under physiological musculoskeletal conditions [17]. Using OVX to mimic osteoporotic conditions, we demonstrated, that loss of SP compromised structural and mechanical bone properties and resulted in accelerated remodeling of hypertrophic callus tissue and reduced hyperalgesia in Tac1-deficient mice (Tac1−/−) during the healing process of intramedullary-stabilized fractures [12].

α-calcitonin gene-related peptide (α-CGRP) is expressed at high density in sensory, trigeminal and vagal ganglia, which project to hypothalamus, thalamus and amygdala, important brain regions integrating nociceptive input. α-CGRP-expressing neurons are polymodal capsaicin-sensitive nociceptors that often co-express SP and noradrenaline [18,19,20]. They are activated by a wide range of noxious stimuli (e.g., thermal or visceral) and contribute to neurogenic inflammation. As the strongest known vasodilator, α-CGRP plays a key role especially in the pathophysiology of migraine, but elevated levels of α-CGRP have also been described for many other pain conditions such as osteoarthritis, gynecological pain, and complex regional pain syndrome [21]. Loss of α-CGRP in knockout mice (α-CGRP−/−) resulted in the development of an age-dependent osteopenia [22]. In vitro, we observed a direct effect on osteoblast and osteoclast metabolism when α-CGRP was absent, as both bone cell types express and autonomously produce the neuropeptide [23].

Noradrenaline (NA), a prominent catecholaminergic neurotransmitter of the sympathetic nervous system (SNS), has a wide range of actions in the body and brain, as it primes for the so-called fight-or-flight reaction. Signaling is mediated via α- or β-adrenergic receptors (AR). NA is an important player in the intrinsic pain control system: on the one hand, NA levels and receptor expression are locally elevated after tissue injury, promoting fiber sprouting, inflammation and neuropathic pain. On the other hand, NA is released from pontine descending anti-nociceptive neurons and can directly modulate, pre- and postsynaptically, the input from primary afferent nociceptors as well as activate inhibitory interneurons in the dorsal horn [24]. NA exerts a versatile role in bone homeostasis and repair; see review [18]. The chemical destruction of the peripheral SNS (sympathectomy, SYX) with 6-Hydroxydopamine (6-OHDA) profoundly impaired mechanical and structural bone properties of fractured and non-fractured bone, and affected callus tissue maturation and pain-related processes as touch sensitivity during fracture healing under physiological conditions [17]. However, it had no effect on nocifensive behavior in an osteoporotic fracture healing model using OVX mice [11].

To date, our results demonstrate a functionally relevant relationship between sensory and sympathetic neurotransmitters, specific healing sequences and pain-related behavior during the healing process of osteoporotic fractures. However, we still lack detailed insights into the modulation of central pain perception and processing in relation to fracture-induced hyperalgesia.

In pain research, i.e., in assessing the involvement of the central nervous system (CNS) in the processing and integration of peripheral signals, functional magnetic resonance imaging (fMRI) is the current gold standard for evaluating brain structure and function in animals and humans. Being non-invasive, multiple measurements can be performed safely in vivo, and the ability to use the same technique in animals and humans renders findings from fMRI highly translational.

Our aim was to characterize the development and resolution of fracture-associated hyperalgesia and fracture healing under impaired musculoskeletal conditions in adult female OVX mice. To this end, we used mouse strains deficient in SP (Tac1−/−) and α-CGRP and mice with non-physiologically low levels of adrenergic neurotransmitters induced by chemical sympathectomy (6-OHDA). For the first time, stimulus-driven and resting-state (RS) fMRI were combined with µCT, biomechanics and neurotransmitter gene expression data [25,26] to assess the influence of SP, α-CGRP and adrenergic neurotransmitters on structural and functional bone properties and fracture-induced hyperalgesia.

## 2. Results

### 2.1. µCT Analysis and Biomechanical Tests

At day 21 post fracture and after the fMRI measurements, bones were prepared for µCT scanning. We analyzed the bond structure and microarchitecture of the bony callus formed at the fracture site plus the trabecular bone properties in the contralateral non-fractured legs. Figure 1 shows representative images of the selected volume of interest (VOI) around the fracture site (Figure 1a) and for trabecular bone analysis in the contralateral legs (Figure 1b).

After µCT analysis, bones were prepared for the biomechanical tests. Two out of four fractured femora in the SYX group and two out of five fractured femora in the WT group failed during the preparation procedure before the application of biomechanical tests (i.e., collapse of diaphysis due to insufficient fracture bridging; see Figure 1c for representative µCT images of the fracture site). Only two fractured femora were included into the SYX group for further analysis. Animals in the WT group were replaced by 3 WT animals of cohort 2 (see Table 4 and Section 4.1.4: Group assignment and dropout). None of the fractured femora within the α-CGRP−/− and Tac1−/− group collapsed during preparation (see Figure 1d for representative µCT images of fracture site).

#### 2.1.1. µCT Analysis

For the fractured femora, no significant difference was detected in trabecular numbers (Tb.N., Figure 2a), trabecular thickness (Tb.Th., Figure 2b) and trabecular separation (Tb.Sp. Figure 2c) in the callus area when comparing all groups.

Total volume (TV) of the calcified tissue around the fracture site was significantly lower in α-CGRP−/− and, by trend, lower in Tac1−/− mice compared to WT (Figure 2d). No significant difference compared to SYX mice was observed presumably due to the high inter-experimental differences of the data points within the SYX group. Bone volume (BV) of the fracture callus was significantly lower at the fracture site of α-CGRP−/− and Tac1−/− compared to SYX mice, whereas no difference was observed between neurotransmitter-deficient mice and WT (Figure 2e). No significant differences were measured regarding the ratio of bone volume/total volume (BV/TV) between all groups (Figure 2f).

At the fracture site, no difference was observed in trabecular bone mineral density (BMD, g/cm^3^) (Appendix A). Likewise, BMD in distal trabecular bone and tissue mineral density (TMD, g/cm^3^) in the cortical bone of the left femora were comparable in all groups (Appendix A).

With respect to the contralateral, non-fractured femora, Tb.N. was significantly higher in the distal part of the leg of α-CGRP−/− and Tac1−/− mice compared to WT (Figure 3a), whereas SYX mice showed no difference compared to WT. Tb.Th. and Tb.Sp. were comparable in all four groups (Figure 3b,c).

TV of trabecular bone in WT, Tac1−/− and SYX mice did not differ significantly, but TV was reduced in α-CGRP−/− compared to WT mice (Figure 3d). BV in the distal femora of Tac1−/− mice was, by trend, increased compared to WT controls. No differences were observed between WT, α-CGRP−/− and SYX mice. Trabecular bone in the distal femur of α-CGRP−/− and Tac1−/− mice revealed a higher BV/TV ratio compared to WT, but not to SYX mice (Figure 3f).

In the contralateral, non-fractured femora we did not observe differences in BMD of the trabecular bone and TMD of the cortical bone (Appendix A).

#### 2.1.2. Biomechanical Tests

No significant differences were detected when shear stress was applied to the fractured femora of all groups (Figure 4a), partly due to the high inter-experimental data variation in the WT group. Further, the SYX group could not be used for statistical comparison due to the low number of biological replicates (*n* = 2). Likewise, shear stress was comparable in the non-fractured femora of all groups (Figure 4b). Torsional stiffness was decreased in fractured femora of α-CGRP−/− compared to Tac1−/− mice. No statistical significance was detected compared to WT mice (Figure 4c). Torsional stiffness of non-fractured femora was comparable between all groups (Figure 4d). After calculation of relative shear stress, no differences were observed (Figure 4e).

### 2.2. fMRI Analysis

#### 2.2.1. Characterization of Neurotransmitter-Deficient Mouse Models

Differential phenotyping of brain activities allowed for assessing how the sensory and sympathetic nervous system influenced basic brain functions in general as well as nociception specifically. Therefore, the baseline (day −2) resting state (RS; Figure 5a) and stimulus-driven networks (left paw, stimulation with noxious temperatures 50 and 55 °C; Figure 5b) were compared between WT animals and each of the three experimental groups.

Regarding the RS networks, α-CGRP−/− mice showed widely reduced FC in the cortex, thalamus, limbic system (diagonal band of broca, hypothalamus, amygdala), brainstem regions (medulla, pons, raphe) and cerebellum. In contrast, processing of nociceptive temperatures was virtually unchanged compared to WT.

Tac1−/− and SYX mice, on the other hand, were both dominated by a comparable prominent increase in RS FC and decrease in nociception-related FC: in RS, Tac1−/− showed vast enhancement of cortical FC, comprising sensory, association and motor cortex. FC was enhanced especially within the cortex as well as between cortex and brainstem, which was also found to a lesser extent in SYX mice. Both groups, Tac1−/− and SYX, showed reduced RS FC within the thalamus, the limbic system (diagonal band of broca, hypothalamus, amygdala) and the brainstem (pons, medulla, raphe, ventral tegmental area), with a stronger overall reduction in SYX mice.

Nociception-related FC was also decreased in both groups within tegmentum, midbrain including periaqueductal grey, lateral and medial thalamus, hippocampus, amygdala, and regions of the reward system (nucleus accumbens, association cortex, diagonal band of broca, septum, olfactory tubercle), being slightly more distinct in Tac1−/− mice.

Gene expression network correlation analysis (Table 1) showed that the differences between the neurotransmitter-deficient mice and WT cannot simply be attributed to the mere deficiency of the respective neurotransmitter, but to complicated interaction between regions, for example, in α-CGRP−/−, only a few differences (in hypothalamus and basalganglia) were found in brain regions expressing α-CGRP, and most changes in FC were found in NA- and SP-expressing regions. Likewise, even though Tac1−/− and SYX lack different transmitters, the differences to WT were highly comparable, and the affected set of brain regions express both NA and SP. Moreover, in SYX mice, depletion of adrenergic neurotransmitters is limited to the periphery, as 6-OHDA cannot cross the blood brain barrier, suggesting peripheral modulation of nociception.

Collectively, loss of adrenergic neurotransmitters as well as SP influences basic brain function and nociception in a comparable manner, whereas loss of α-CGRP leads to a differential effect and has no direct impact on baseline nociception. These findings are backed up by the overall behavior of the mice, with α-CGRP−/− being quite docile and friendly, whereas Tac1−/− and SYX display a more aggressive-defensive behavior when being handled.

#### 2.2.2. Nociceptive Functional Connectivity

To analyze the temporal development of hyperalgesia induced by the fractured leg, the brain’s response to phasic noxious heat stimulation of the ipsilateral hind paw (50 and 55 °C) was analyzed separately for all the experimental groups (Figure 6). For that purpose, FC-based brain networks of noxious stimulation of the fractured site at day 5 and day 21 past fracture were compared to baseline FC at day 2 before fracture.

WT mice showed enhanced FC between contralateral (to the fracture) amygdala and hippocampus as well as reduced cortical, hippocampal and thalamic FC at day 5. At day 21, reduction in FC was abolished, but a very specific bilateral enhancement of hippocampal FC had developed.

α-CGRP−/− mice also displayed enhanced amygdala and hippocampal FC at day 5, but in this case, bilateral. The cortical and thalamic reduction of FC found here was more pronounced than in WT. FC patterns at day 21 were dominated by a vast decrease in FC in thalamus and midbrain (esp. periaqueductal grey (PAG)), that was highly specific for α-CGRP−/−.

Tac1−/− mice showed also the bilateral increase within the amygdala and hippocampus, and a slight decrease only in cortex. Here, the increased FC of amygdala and hippocampus remained throughout day 21, albeit only on the ipsilateral side. Slightly reduced FC was found for cortex and thalamus at day 21.

SYX displayed a vast increase in FC within cortex (esp. association but also sensory and motor cortex), thalamus, hippocampus and the midbrain (including PAG) and the reward system at day 5, and a small decrease in some cortical and cerebellar FC. The prominent increase was resolved almost completely by day 21 except for association cortex and hippocampus. The decrease in cortical and cerebellar FC was stable throughout 21 days.

Notable in all four groups, FC at day 5 was enhanced in the amygdala (partially also involving hippocampus), when hyperalgesia is high. Of the neurotransmitters analyzed in the gene expression network correlation analysis (Table 2), SP was the only transmitter that modulated amygdala (as well as hippocampus, though this region is also affected by noradrenaline). However, only in Tac1−/−, mice (no SP), the enhanced amygdala–hippocampus FC persisted to day 21, indicating reduced control of amygdala–hippocampus crosstalk.

#### 2.2.3. Resting-State Functional Connectivity

RS networks are known to be altered by many pathological conditions, and even relatively short-term events such as acute nociceptive fMRI sessions (publication in progress). By comparing RS networks of days 5 and 21 to baseline RS FC, we aimed to evaluate a possible constant, long-lasting influence of fracture-evoked hyperalgesia on RS networks (Figure 7).

Of the four mouse groups, WT mice exhibited the overall weakest changes. These mice showed enhanced FC in ipsilateral sensory cortex and reduced FC within cortex and limbic system at day 5, when hyperalgesia is thought to be high. At day 21, cortical enhancement was reduced, but the decrease in cortex and limbic system was even slightly pronounced.

A-CGRP−/− mice showed stronger ipsilateral cortical enhancement at day 5 and additionally enhanced FC in the limbic system and brainstem. At day 21, FC enhancement in these structures was even more pronounced. Of note, α-CGRP−/− mice were the only group in which the increased cortical FC persisted significantly until day 21. Decreased FC was minimal and comparable at days 5 and 21, scattered throughout the brain.

Tac1−/− mice displayed a strictly focused increase in ipsilateral cortical FC but the strongest reduction of cortical-brainstem FC at day 5. Both findings were resolved mostly at day 21; however, the decreased brainstem FC partly remained. Interestingly, Tac1−/− mice constituted the only group with clear lateralization effects: at day 5, the left subcortex showed reduced FC compared to the right cortex and right subcortex enhanced FC compared to left cortex. This strong lateralization subsided to day 21, especially regarding reduced FC.

SYX mice also showed focused ipsilateral cortical enhancement at day 5, comparable to that of WT. Decreased FC however resembled more the Tac1−/− phenotype, although a bit weaker, less distinct and with stronger involvement of association cortex. Until day 21, both findings resolved to be mostly comparable to the WT level.

Together, fracture-induced hyperalgesia was found to leave typical imprints on RS FC, regardless of the mouse model. Key features were enhanced ipsilateral FC, especially of the sensory cortex, and partially brainstem regions (mainly medulla). Tac1−/− mice showed a distinct phenotype with the strongest and also unilateral decrease in FC compared to WT.

According to the gene expression network correlation analysis (Table 3), the neurotransmitter NA represented modulation of cortex and brainstem best, as both the cortex and brainstem contain noradrenergic neurons. SP also has receptors on brainstem regions, but does not modulate sensory cortex.

In the experimental groups, with the exception of the α-CGRP−/− group, the findings from day 5, resembling high hyperalgesia, resolved to a great extent at day 21, the time point of the bone remodeling phase. The least difference to the baseline was found in WT, whereas Tac1−/− and SYX mice showed a persisting reduction in brainstem FC up to day 21. In the α-CGRP−/− group, the differences to the baseline were even stronger at day 21.

## 3. Discussion

Chronic pain after osteoporotic fractures is a known problem [10]. Understanding the link between the sensory and sympathetic nervous system and fracture-induced hyperalgesia in a frequently occurring pathological condition as osteoporosis can provide deeper knowledge of the process of pain chronification, which thus may lead to treatment optimization for osteoporotic bone fracture. The aim was to investigate the relationship between sensory and sympathetic signaling, fracture healing and fracture-induced hyperalgesia with focus on the CNS by stimulus-driven as well as resting-state fMRI supported by structural µCT and assessment of femur biomechanical properties. The present study is based on an externally stabilized femoral fracture model after the surgical induction of osteoporosis by OVX in well-established mouse models with deficits in sensory and sympathetic neurotransmitters. OVX was performed in 10-week-old mice and is an approved model to mimic post-menopausal osteoporosis [1,27], and the chosen time span of four weeks is acknowledged to be sufficient to induce osteoporotic changes in bone metabolism and structure [5,28].

### 3.1. Opposite Effects of Sensory and Sympathetic Signaling on Bony Callus Microarchitecture in the Remodeling Phase

Systemic peripheral 6-OHDA treatment results in an 80% reduction of peripheral adrenaline and NA of which the remaining neurotransmitters then primarily act via high-affinity α-adrenergic receptors (AR) [8,29]. The hereby-induced modulation of peripheral sympathetic signaling in OVX mice resulted in altered microstructural properties of the bony callus which were partly opposite compared to Tac1−/− and α-CGRP−/− mice. Chemical sympathectomy increased BV in this mouse group, whereas no differences were observed compared to WT. These findings contrast previous data showing no differences in trabecular structure, BV, TV or BV/TV between the bony callus of SYX and Tac1−/− mice 21 days after fracture [17]. Notably, in the previous study, male mice were used with intramedullary-stabilized fractures, while the present study was conducted using OVX female mice with external fracture fixation. Therefore, hormonal differences—with OVX mice reacting more sensitively to bone traumata and subsequent bone regeneration— and differences in fracture stabilization might account for the opposing results. Another study reported that 6-OHDA treatment in a mouse transverse fracture model impaired bone micro-architecture at the fracture site at two and four weeks after fracture compared to the control group [30]. A comparable effect of OVX on trabecular BV/TV in WT- and global β-AR-deficient mice was demonstrated in a study of Bouxsein and colleagues, in which the functional α-AR binds catecholamines, i.e., NA, with higher affinity resulting in different cellular responses [31]. Our data let us assume a compensating effect on bone remodeling due to estrogen hormone deficiency in the absence of sympathetic signaling at the fracture site. Nevertheless, differences in fracture stabilization (intramedullary vs. external stabilization) presumably critically contribute to the contrasting data sets.

In general, the absence of a variety of neurotransmitters from sympathetic nerve fibers seems to affect late-stage fracture healing under pathophysiological musculoskeletal conditions, possibly more when compared to loss of sensory neuropeptides. The delay in bony callus maturation after intramedullary-stabilized femoral fractures demonstrated in ovariectomized SYX mice supports this hypothesis [12]. However, estrogen deficiency in combination with reduced adrenergic signaling might compensate effects of altered adrenergic signaling along the callus maturation timeline.

Ding et al. reported impaired fracture healing in OVX mice, accompanied by a drastic decrease in SP in the callus tissue from week one to eight during callus maturation, concluding a possible regulatory role of SP during osteoporotic fracture healing [32]. This study revealed that loss of SP reduced the TV by trend at the fracture site of Tac1−/− mice but otherwise had no further effects on microarchitecture of the bony callus. This is in line with our previous observations, demonstrating only minor effects of loss of SP on bony callus remodeling during late-stage fracture healing. However, global SP deficiency resulted in strongly impaired bone structural properties in the proximal trabecular bone of the non-fractured femora compared to WT under physiological conditions [12,17]. In this previous study, we suggested a physiological effect of SP on bone remodeling and bone growth already right after birth.

Interestingly, OVX resulted in a more severe decrease in bone quality in the distal trabecular bone of the contralateral non-fractured femur of WT compared to Tac1−/− mice, demonstrated by lower Tb.N., BV and BV/TV. Our data therefore suggest a slightly protective effect of SP deficiency on OVX-induced bone loss. This effect was not observed in the proximal trabecular bone of the tibia after blocking SP signaling in OVX mice using the Neurokinin 1 Receptor (NK1R) antagonist L-703606 [33]: Zheng and colleagues measured similar Tb.N., Tb.Th., Tb.Sp. and BV/TV in OVX mice with antagonist treatment and vehicle controls. In a subsequent study, they further reported that adverse effects in trabecular structural bone properties of tibiae were aggravated by this NK1R antagonist in OVX mice, demonstrated by reduced BV/TV, Tb.N. and Tb.Th. and increased Tb.Sp. [34]. Various reasons might account for the conflicting results. Zheng et al. started with the NK1R signaling blockade three weeks after OVX, probably influencing OVX-induced bone remodeling, whereas the global SP deficiency of our mice modulate overall bone homeostasis, including already embryonal developmental processes. In addition, besides NK1R, SP can bind to NK2R and NK3R [35,36], which are not affected by the blockade and might modulate bone remodeling differently compared to a global neuropeptide deficiency. We assume that a global SP deficiency modulates bone growth and remodeling with only minor influence specifically on bony callus remodeling during fracture healing under both pathophysiological and physiological musculoskeletal conditions.

Loss of the sensory neuropeptide α-CGRP strongly impacts on bone homeostasis, demonstrated by the development of an osteopenic bone phenotype in six-month-old α-CGRP-deficient mice [22]. The age-dependent bone loss was supported by an in vitro study of our group, showing that a global α-CGRP deficiency affected bone cells of three and nine-month-old mice differently with respect to bone resorption and bone formation parameters [23]. In the present study, OVX had less severe effects on bone structural properties in α-CGRP-deficient mice compared to WT mice. Although TV was reduced in the bony callus of α-CGRP−/− mice, no difference was observed with respect to BV/TV or trabecular parameters. Further, Tb.N. and BV/TV were increased but TV was reduced in the contralateral femur of α-CGRP−/− mice. The effect might be due to the younger age of mice, used in this study, as the osteopenic bone phenotype was reported in 6-month-old mice [22]. In addition, changes in neurotransmitter quantity after OVX might account for the differences, as Xie et al. reported reduced SP, CGRP and VIP staining in the tibia, but increased neuropeptide signals in the dorsal root ganglia (DRG) of OVX rats [37]. Total absence of α-CGRP might shift sensory neuropeptide balance and bone metabolism, and consequently, these mice react differently to OVX-induced modulations. Contradictory results were demonstrated by a study from Appelt et al., reporting an insufficient callus formation and bridging in α-CGRP-deficient mice [38]. This supports our hypothesis, that OVX-induced hormone deficiency probably influences multiple neuronal and inflammatory pathways, thereby negatively affecting bone remodeling, resulting in insufficient healing outcomes.

### 3.2. Impaired Bridging at the Fracture Site in WT and SYX Mice after OVX

Biomechanical tests of fractured and non-fractured femora revealed no significant differences between WT and neuropeptide-deficient mice after OVX. In earlier findings, torque and stiffness were reduced in male Tac1−/− and SYX compared to WT mice under physiological conditions [17]. According to those data, the effects in Tac1−/− mice might be linked to the altered hormonal status plus SP deficiency.

Limitations: The data from OVX SYX mice were unusable for statistical analysis due to pre-analytical collapse of the diaphysis at the fracture site while preparing the legs for the tests. Similarly, two fractured femora of the WT group were useless. Bony callus scans of these femora showed insufficient bridging of the fracture gap, which was likely the reason for the pre-analytical collapse. This was not observed within the Tac1−/− and α-CGRP−/− groups where complete callus bridging was visible in all the fractured femora. Nevertheless, sufficient callus bridging was observed for the remaining SYX and WT animals. These data further push our suggestion, that fracture healing under pathophysiological musculoskeletal conditions is more affected after sympathectomy and that OVX-induced changes on bone metabolism are more pronounced in WT than in sensory neuropeptide-deficient mice.

### 3.3. Depletion of Sensory and Sympathetic Neurotransmitter Signaling Modulates Interaction of Brain Structures

FMRI measurements were not performed with the mouse groups of the present study before OVX. To address the influence of OVX on the female mice at least partially, we compared WT-OVX (four weeks after OVX, before fracture) from this study with non-OVX C57Bl/6 females from a different fMRI study. No significant differences in terms of network structure and function were found (Appendix A).

Whereas the role of sensory and sympathetic neurotransmitters in nociception is well described on the molecular and behavioral level [39,40], less is known about impact on interactive brain function in general, and nociceptive cerebral processing in particular. Moreover, the link between the sensory and sympathetic nervous system and fracture-related nociceptive processing in the brain has not yet been examined. Therefore, by comparing baseline brain FC of WT mice to the different neurotransmitter-deficient mouse models used in this study, we first investigated basic FC differences in RS and nociceptive processing. All three experimental groups differed significantly from WT, demonstrating that sensory and sympathetic signaling influences cerebral processing in general, and nociception in particular, through altered input from the periphery and/or central modulation.

Interestingly, changes found in Tac1−/− and SYX mice were highly comparable regarding both resting state and nociception. One has to consider that there might be a compensation of the absence of SP through hemokinin-1 (HK-1)—derived from the tachykinin-4 gene—which is able to activate the NK1R [41]. Hunyady et al. reported a pro-nociceptive role of HK-1, via NK1 receptor activation in acute murine inflammation models, which is different from SP-mediated actions.

Additionally, our previous study assessing nightly locomotion in the home cage has proven that Tac1−/− and SYX mice both showed similar behavior in the form of reduced movement after intramedullary-stabilized fractures [12]. These findings suggest that SP and adrenergic sympathetic neurotransmitters both play a vital and comparable role in general brain function which may be due to direct interaction or convergence of ascending neurons. Interaction of the two NT is well known, for example, NA is able to trigger the release of SP in rat dorsal root ganglia and SP and adrenaline interplay directly regulates blood pressure [42].

Tac1−/− and SYX mice showed enhanced RS FC in sensory, association and motor cortex, as well as between cortex and brainstem, regions involved in the localization and evaluation of stimuli and internal processes. Decreased RS FC was found for the thalamus, limbic system and within brainstem regions involved in the filtering of incoming information, emotional processing and body homeostasis but also partially involved in descending pain pathways (PAG, raphe, ventral tegmental area and medulla). Regarding nociceptive FC, both groups showed reduced FC of tegmentum, midbrain (including PAG), lateral and medial thalamus, hippocampus, amygdala, and the reward system, regions involved in the filtering and relay of incoming sensory signals, memory formation, evaluation of intrinsic incentives and body homeostasis as well as descending pain pathways and behavioral responses to noxious stimuli. Interestingly, sensory-cortical FC was relatively unaffected, indicating unchanged valence of the nociceptive stimuli. With respect to present µCT results and data from our previous studies, where we could demonstrate partly contradicting effects of the sympathetic and sensory nervous system on fracture healing and bone metabolism, fMRI data point to an opposing role of sympathetic and SP-related activity in the periphery compared to brain function: on the one hand, SP is a potent pro-algesic mediator in the body periphery, being released by activated nociceptors during inflammatory processes. On the other hand, SP release in the ventral tegmental area was found to induce analgesia by engaging midbrain dopaminergic neurons, likely part of the stress-induced analgesic pathways [43]. A similar ambivalent mechanism is known for NA (see introduction).

In contrast, compared to WT, loss of α-CGRP had no influence on baseline nociception, but decreased RS FC globally in the cortex, thalamus, limbic system, brainstem and cerebellum. With α-CGRP being a powerful vasodilator, this might be attributed to a possible minor baseline hypo-perfusion of the brain, which is overwritten by the neurovascular coupling that actively triggers cerebral blood flow when performing a task, such as nociceptive processing. Contrarily, α-CGRP-antagonists as well as α-CGRP itself were shown not to influence cerebral hemodynamics in principle [44,45]. RS regions affected by knockout of α-CGRP are—besides many other tasks—also involved in the sensory-descriptive motor, as well as affective aspects of nociception, probably setting the basis for deviating responses to noxious thermal stimulation after fracture.

Opposing to the comparable effects of SP and α-CGRP on bone structural parameters, being often co-expressed on primary afferent neurons and both being powerful vasodilators, they seem to influence brain functional connectivity differently. The notion that Tac1−/− and SYX mice both have functional α-CGRP signaling and present comparable brain FC that differs markedly from that of α-CGRP−/−, points to an important role of this neuropeptide on interactive brain function after OVX. Contrasting actions of SP and α-CGRP can also be seen in migraine therapy: even though both peptides promote the typical neurogenic inflammation, blockade of SP signaling through application of NK1R antagonists has no effect on acute migraine [46], whereas blockade of α-CGRP or its receptor has a strong potential for acute and preventive treatment of migraine-related pain [47].

### 3.4. Bone Fracture Evokes Longitudinal Changes in Nociceptive Processing

Bone fracture evoked changes in nociceptive processing and RS FC, most likely attributed to fracture-associated hyperalgesia. Regarding nociception, the main characteristic found in all groups was enhanced amygdala-hippocampal FC at day five, which resolved mainly until day 21, except in the Tac1−/− group. The central lateral amygdala has long been known as a key region of fear processing [48] and fear memory [49], and has recently been shown to play an important role in nociceptive modulation [50,51]. With Tac1−/− mice globally lacking the neurotransmitter SP, and amygdala expressing the NK1R (being a target region of SP), the enhanced amygdala–hippocampal FC found day 21 only in this group may underline a possible control effect of the amygdala–hippocampus crosstalk by SP. Interaction between the amygdala and hippocampus is modulated by chronic stress [52], and blockade of the Tachykinin1-receptor NK1 in stress and anxiety-related brain regions including the amygdala has positive effects on anxiety and negative mood [53]. These findings underline a possibly SP-mediated lasting imprint of the fracture-associated hyperalgesia on mood and memory-associated brain regions as amygdala and hippocampus. The central amygdala was also found to mediate thermal hyperalgesia evoked by alcohol withdrawal in dependent rats via modulation of GABAergic input to the PAG [54]. The PAG, a key effector region of the descending pain modulatory pathway [55], mediates strong analgesia via endogenous cannabinoids and opioids [56,57]. Interestingly, α-CGRP−/− mice revealed a unique decrease in thalamic-midbrain-diagonal band FC, that was notable at day five and was pronounced at day 21, here with a clear involvement of the PAG. In animals and humans, the PAG has direct afferents to sensory thalamic nuclei, possibly modulating nociception also in an ascending manner. The findings in the α-CGRP−/− group may therefore indicate reduced thalamic filtering of incoming nociceptive signals and/or impaired PAG ascending and descending modulation thereof. Our previous behavioral data showed increased and prolonged mechanical hyperalgesia (reduced weight at withdrawal) in OVX α-CGRP−/− mice, where the detection of increased SP levels in the serum of α-CGRP−/− mice was found as a possible counter mechanism [12]. Collectively, although α-CGRP−/− mice did not differ from WT mice in baseline nociception, they exhibited the most striking changes in fracture-related nociceptive processing compared to the other groups, and these findings support the notion of increased hyperalgesia in the absence of α-CGRP.

### 3.5. Bone Fracture Induces Recognizable Changes on Resting-State Functional Connectivity

The analysis of BOLD signal fluctuations at rest in humans had revealed a consistent set of stable, state-dependently activated brain networks, such as the default mode, visual, motor and executive networks [58]. Equivalent networks could be identified in animals, e.g., mice [59], even though the factor of anesthesia needs to be taken into account [60]. To allow comparisons between groups, the same anesthetic regimen needs to be applied to all animals. In the study presented here, low-dose isoflurane anesthesia was used, which is known to impact brain glucose metabolism and blood flow in a dose-dependent manner [61,62]. However, low-dose isoflurane was found to yield the most similar results in RS FC to awake animals, in contrast to α-chloralose and urethane [63]. RS networks offer high sensitivity to detect even slight changes in the interaction of the brain region. They show aberrations in many (psychiatric, developmental and neurodegenerative) disorders [64,65,66], including attention deficit hyperactivity disorder, autism, schizophrenia, Alzheimer’s disease and Parkinson’s disease. This raises hope that RS fMRI might be useful for the discovery of biomarkers that allow the early identification and survey of disease progression and therapeutic intervention.

Here, analysis of RS connectivity identified enhanced cortical but reduced cortex-brainstem connectivity as a hallmark of fracture-associated pathological conditions, most likely hyperalgesia, in all four groups of mice. Even though RS FC is task-independent, any profound changes in the interaction of brain regions, e.g., induced by a pain condition, also leave footprints in RS FC. Therefore, RS changes in nociception-related brain regions were not unexpected.

If WT mice were set as reference for a physiological healing process, results suggest that α-CGRP−/− mice showed enhanced hyperalgesia at day five, and prolonged hyperalgesia at day 21, with persistently enhanced cortical FC, reflecting the enhanced mechanical hyperalgesia, described in our previous study [12]. The α-CGRP−/− group patterns suggest that the resolution of hyperalgesia may be impaired by the deletion of α-CGRP, probably due to the resulting elevation in SP levels detected in our previous study. In contrast, results from Tac1−/− and SYX mice were more comparable to WT mice, with Tac1−/− mice showing the strongest and most focused cortical enhancement at day five. Interestingly, cortical enhancement was focused to the left hemisphere, ipsilateral to the fracture, whereas changes in cortex-brainstem FC were mainly bilateral. In the Tac1−/− group, cortical connectivity also showed clear lateralization effects: enhancement between right subcortex and left cortex and reduction between left subcortex and right cortex. This effect was prominent at day five and waned to day 21.

On the one hand, a reduction in somatosensory cortical FC is usually described for (chronic) pain in task-based fMRI, and the localization is usually contralateral to the site of the chronic stimulus (in our case fractured left femur) [67,68,69]. On the other hand, lateralization was described for many RS networks, with the sensorimotor network showing a greater magnitude of lateralization towards the left side in humans [70]. Furthermore, resting-state lateralization effects were previously described for other focused unilateral conditions, such as epilepsy [71,72], tinnitus [73] or migraine [74], concluding that lateral events can lead to lateral imprint on RS FC. Cortical regions encompassed not only the somatosensory, but also associative structures (cingulate, frontal, orbital and prelimbic cortex). These regions play an important role in the evaluation of pain amplitude, aversion and also modulation, e.g., by reducing pain and aversion ratings via distraction and expectancy [75,76]. Supporting our findings, RS FC analysis of rheumatoid arthritis patients revealed enhanced frontal-sensorimotor FC [77]. Flodin et al. attribute those findings to enhanced affective processing in chronic pain patients, a conclusion that supports our findings [77].

The reduction in brainstem FC concerned mainly regions of the medulla, such as nucleus of solitary tract, gigantocellular nucleus, parvicellular reticular nucleus and area postrema, regions usually involved in body homeostasis tasks and autonomous movements, such as breathing and swallowing, but some are also an important part of the descending pain modulatory pathway. The PAG, receiving afferents from anterior cingulate cortex, sends projections into the rostral ventral medulla, which in turn have inhibitory afferents directly to the dorsal horn [78,79]. The nucleus of solitary tract processes not only gustatory and visceral input but also ascending as well as descending nociceptive information. The decreased FC between cortex and medulla found especially in Tac1−/− and SYX mice may indicate involvement of the descending pain modulation in the fracture healing process, as changes are most prominent at day five, but persist—although weaker—to day 21. Further, this may account for the reduced mechanical hyperalgesia in Tac1−/− and SYX mice shortly after fractures, described in our previous studies [12,17]. In both studies, baseline locomotion was similar in male mice and female + OVX mice when comparing the respective groups of WT, Tac1−/− and SYX mice [12,17], indicating, that the results shown here can be attributed to fracture healing specifically under OVX conditions.

### 3.6. Changes in Neurotransmitter Abundancy Does Not Linearly Account for the Modulatory Effect on Brain Functional Connectivity

By analyzing gene expression data from the Allen Brain Institute, we tried to attribute the changes found in nociception-related FC and the more general RS FC directly to the abundance of the neurotransmitters α-CGRP, SP and NA. Interestingly, despite being expressed in the brain, deficiencies of α-CGRP and SP were found to act primarily in the periphery modulating the fracture healing process and altering the ascending input, instead in a central, more direct way: neurotransmitter-specific brain networks (i.e., brain structures expressing the NT or receptors) showed no critical differences in FC between the four experimental groups. NA, which is only depleted in the periphery by 6-OHDA but not in the brain (SYX group), is a likely candidate for the striking modulation of cortical-brainstem FC found in RS, as there is a high overlap between NA and AR expression and those brain regions.

### 3.7. Summary

In sum, using female OVX mice as a model for post-menopausal osteoporotic conditions, bone microarchitecture was more affected in the absence of sympathetic neurotransmitters than after loss of the sensory neuropeptides SP and α-CGRP. Furthermore, the hormonal status appears to play a role: OVX seems to have a stronger effect on the bone homeostasis of female WT animals, resulting in stronger deterioration of structural bone properties than in female Tac1−/− and α-CGRP−/− mice.

Whereas SP- and α-CGRP-deficiency led to comparable effects on bone structural and functional properties with respect to µCT- and biomechanical data, loss of α-CGRP affects brain FC differently compared to SP deficiency and sympathectomy. Nonetheless, all experimental groups differed from WT concerning the longitudinal survey of stimulus-driven and RS FC.

Fracture of the left femur induced focal changes in brain structure interaction identifiable in stimulus-driven (amygdala, hippocampus) and RS (cortex, brainstem) FC. In both cases, changes were (with exception of α-CGRP−/−) most prominent at day five, supporting the idea that these modulations reflect fracture-induced short-term hyperalgesia. However, persisting amygdala–hippocampus involvement at day 21 found in the stimulus-driven fMRI in WT and Tac1−/− mice suggest lasting imprint on emotion–memory-related key regions, or even persisting central sensitization processes.

Striking hallmarks of RS FC were enhanced cortical but reduced cortex-medulla connectivity, likely evoked by fracture-induced hyperalgesia.

Fracture healing in α-CGRP−/− mice, in contrast to WT, Tac1−/− and SYX mice, demonstrated enhanced and prolonged mechanical hyperalgesia reflected in a lasting impact on RS FC, suggesting that the presence of α-CGRP is crucial for the timely resolution of hyperalgesia.

## 4. Materials and Methods

### 4.1. Animals

#### 4.1.1. Animal Description

All experiments were conducted in accordance with the local veterinary administration (Umweltamt, Dept. Veterinärwesen und Verbraucherschutz, Regensburg, approved on 19 December 2014) and in agreement with the local authority (Regierung Unterfranken, Bayern, Germany) controlling animal experimental usage (Az.: 54-2532.1-23/14). Female mice (for description, see Section 4.1.2 on mouse models) at the age of 10 weeks [1,2,5] were subjected to ovariectomy (OVX) to induce an osteoporotic phenotype (see Section 4.2.2 for surgical procedures and Table 4 for sample sizes).

#### 4.1.2. Mouse Models

The following mouse models were used for the study: α-CGRP-deficient (α-CGRP−/−) mice were generated by an insertion of a stop-codon into the α-CGRP coding region [7] and were a generous gift from Ronald B. Emeson (Department of Pharmacology, Molecular Physiology, and Biophysics, Vanderbilt University School of Medicine, Nashville, TN, USA). SP-deficient (Tac1−/−) mice were generated by Zimmer et al. [6]; they do not produce SP due to a targeted mutation in the tachykinin 1 gene. Both strains were bred on a C57Bl/6J background in the central animal facility of the University of Regensburg. C57Bl/6J wild type (WT) mice were obtained either directly from Charles River Laboratories (Sulzfeld, Germany) or partially as breeding offspring (breeding pairs: originated from Charles River) from the central animal facilities of the University of Regensburg. For sympathectomy (SYX), WT mice were injected i.p. four times with 80 mg/kg 6-hydroxydopamine (6-OHDA) dissolved in PBS and stabilized with 0.1% ascorbic acid. Injections were performed on days -8, -7 and -6 before and on day 14 after fracture to erase sympathetic peripheral nerve endings resulting in an 80% reduction of adrenergic neurotransmitter supply [8,80]. All mice of the WT, α-CGRP−/− and Tac1−/− groups received i.p. 150 µL PBS with 0.1% ascorbic acid as sham treatment according to this schedule.

#### 4.1.3. Animal Housing

Mice were housed at 21 °C in a 12/12-h light/dark cycle in groups of 2–4 mice at the animal facility of Experimental Pharmacology, University of Erlangen-Nuremberg. After fracture induction, animals had to be single-housed, to prevent manipulation of the external fixator, and therefore perturbation of fracture healing by cage mates. Food (Ssniff Maintenance, Soest, Germany) and water were always provided ad libitum.

#### 4.1.4. Group Assignment and Dropout

Wild type mice were randomly assigned to the WT or SYX group. One mouse of the SYX group died after the first fMRI measurement and was therefore excluded from further analysis.

The fractured femora of two out of four SYX and two out of five WT animals collapsed (i.e., breakdown of diaphysis at fracture site) during preparation for biomechanical tests. Fractured femora of SYX animals could not be replaced and had to be excluded from the fracture SYX group. Non-fractured right femora of remaining four SYX mice were used for the biomechanical tests. Three WT animals of study cohort 2 (see Table 4) were used as replacement in the biomechanical tests.

### 4.2. Protocols and Analysis Overview

#### 4.2.1. Study Design and fMRI Paradigm

To induce an osteoporotic phenotype, both ovaries were removed (see surgical procedures) 28 days before fracture setting (see Appendix A). Mice of the SYX group (sympathectomized mice) were treated with i.p. 6-OHDA at days −8, −7 and −6 before fracture and day 14 after fracture. All other mice were sham-treated with PBS (see mouse models). Two days before fracture induction (see Section 4.2.2 on surgical procedures), mice were subjected to fMRI for determining the baseline state and again five days after fracture, when fracture-induced hyperalgesia was thought to be maximal. The last fMRI measurement was performed 21 days after fracture, when hyperalgesia was thought to be low [81]. Exact fMRI procedures and stimulation are described in Section 4.2.5 on the fMRI preparation, protocols and stimulation paradigm (see also Appendix A).

#### 4.2.2. Surgical Procedures

For the ovariectomy (OVX), female mice at the age of 10 weeks were anesthetized using a mixture of 0.5 mg/kg medetomidin hydrochloride (Dorbene^®^, Dr. E. Graeub AG, Bern, Switzerland), 5 mg/kg midazolam (Ratiopharm, Ulm, Germany) and 0.05 mg/kg fentanyl (Janssen, Neuss, Germany) and were placed on a temperature-controlled heating pad to maintain body temperature during operation. Eyes were covered with an eye ointment (Bepanthen^®^, Bayer, Leverkusen, Germany) to prevent exsiccation damage. Bilateral OVX was conducted by ligation and removal of both ovaries [11,27]. Buprenorphine hydrochloride (0.1 mg/kg; Buprenovet^®^, Elanco, Greenfield, IN, USA) was injected s.c. for analgesia directly after OVX, followed by antagonization of the anesthesia with 2.5 mg/kg Atipamezol (Antisedan^®^, Orion Corporation, Espoo, Finland) and 0.5 mg/kg Flumazenil (Hexal, Holzkirchen, Germany). After 28 days post-OVX surgery, femoral osteotomies were set using a standardized mid-diaphyseal osteotomy-fracture model [82], based on an external fixation device (EXTERNAL FIXATION—MouseExFix, RISystem AG, Landquart, Switzerland). Shortly, after anesthesia (similar to OVX surgery), the femur was exposed by penetration of the fascia latae between the gluteus superficialis and the biceps femoris muscles. The fixator block was placed in a cranio–lateral position using four screws (mini-Schanz screws). The first screw was adjusted proximal, the second distal of the trochanter tertius. After placing the third and fourth screw, a Gigli saw (0.45 mm) was used to create a 0.5 mm osteotomy (in the further referred to as ‘fracture’) in the middle of the diaphysis, between the third and fourth screw. Finally, the wound was closed by suturing the muscle tissue with an absorbable thread, followed by closing the skin using a non-absorbable thread. Buprenorphine hydrochloride (0.1 mg/kg, s.c.) was given directly after the procedure for post-operative analgesia before antagonizing anesthesia as described above. Afterwards, buprenorphine was applied twice a day up to and including the third post-operative day.

#### 4.2.3. µCT Protocols and Analysis

Mice were euthanized with a CO_2_ overdose on day 21 after fracture setting and fractured as well as non-fractured legs were prepared for µCT analysis. Therefore, skin, muscle tissue and the external fracture device were carefully removed from the femoral bones. Samples were stored at −80 °C until µCT analysis was conducted. For the fractured femora, the micro-computed tomography system (Röntgenprüfsystem v|tome|x s 240 Research/Edition V2.5, GE Sensing & Inspection Technologies GmbH, Wunstorf, Germany, DFG number: INST 102/11-1 FUGG) was used to scan the frozen samples with a voxel size of 6.5 µm. The following scanning parameters were used: X-ray tube was operated at 50 kV and 500 µA; an integration time of 333 ms and 1800 images/360° were used. Data reconstruction for analysis of structural parameters was performed using automatic geometry calibration without using further filters.

Directly before the biomechanical tests—to evaluate the bone diameter—the micro-computed tomography system (Skyscan 1176, Bruker, Kontich, Belgium) was used to scan both the non-fractured and fractured frozen samples with a voxel size of 8.9 µm. The x-ray tube was operated at 45 kV and 555 µA with a 0.2 mm Al filter. Every 0.5° the average of 5 frames with an integration time of 1100 ms were taken for a 180° sweep. Data reconstruction was performed using the included NReconSoftware, with adaptive misalignment compensation, 8-bit greyscale and neither smoothing filters nor beam hardening and ring artifact reduction. The phantom of Bruker, provided with the Skyscan 1176 device, was used for measurement of bone density.

Bruker’s DataViewer and CTAn software were used for analysis of bone microarchitecture of the fracture callus. For the fractured femora, the first scans with higher resolution of 6.5 µm were used. For the non-fractured femora, the scans obtained before the biomechanical tests were used for analysis. To analyze the volume of interest (VOI), the top and the bottom of the VOI were set 200 layers above and below the midline of the fracture site, spanning a total VOI of 400 layers (Figure 2). The global threshold with shrink-wrap method (Stretch over holes; diameter 100 pixels) was used to wrap the cross-sectional VOI border around the callus region as described in Bruker method note MN116. Trabecular number (Tb.N.), thickness (Tb.Th.) and separation (Tb.Sp.) and total tissue volume (TV), bone volume (BV) and bone volume/total volume (BV/TV) were analyzed with Bruker’s CTAn software. For measuring bone mineral density (BMD, g/cm^3^), a range of 85 layers of trabecular bone was analyzed 15 layers beneath the growth plate using the shrink-wrap method to wrap the VOI border trabecular region of interest. BMD of trabeculae in the fracture callus was measured in a VOI spanning the distance between the fractured cortical bone ends. Tissue mineral density of the cortical bone (TMD, g/cm^3^) was measured over a VOI of 85 layers starting 100 layers beneath the growth plate.

#### 4.2.4. Biomechanical Analysis

Non-fractured right femora and fractured left femora were used for biomechanical tests at time point 21 days after setting the fractures. Prior to biomechanical testing, frozen legs were thawed overnight at 4 °C. External fixation was already removed from the fractured femora for the µCT analysis. Some µCT scans were obtained to evaluate the bone diameter (see Section 4.3.1 on µCT protocols and analysis). Non-fractured and fractured femora were placed into a vertical position in a clamping slide, cartilage covered surface of condyles were placed as anterior. The distal and proximal ends were cast into three compound cyanate resin (Rencast FC53, Huntsman, Basel Switzerland) at a ratio of 1 part cyanate, 1 part polyol and 3 parts Al(OH)3-filler. As a holding fixture for the cast, custom 3D-printed adaptors were created out of polylactic acid (PLA). The positioned femora were placed into a LM1 test machine (TA Instruments/ElectroForce, New Castle, USA) with a linear and torsional actuator. A compression preload of 1N was applied. After reaching the preload, the actuator started rotational movement at a speed of 0.5°/s. Compression was measured using the included 22.2 N load cell (calibrated to <0.5% error) and torque until failure was measured with a torque load cell 200 Nmm (calibrated to <0.5% error). In addition to the torque to failure, the angle to failure was obtained by read-out of the torsional actuator. The data analysis was performed with a customized Matlab script (R2019b) calculating the shear stress under the assumption of an ovoid rod using the µCT-obtained dimensional data. To compare the biomechanical quality of newly formed bone with the existing bone, the results of fractured femora for each animal were normalized to the results of contralateral non-fractured femora and shown as percentage [contralateral non-fractured = 100%].

#### 4.2.5. FMRI Preparation, Protocols and Stimulation Paradigm

Mice were initially anesthetized with 5% Isoflurane in medical air for 4 min, before being transferred to a ventral position onto a specially designed acrylic cradle. The cradle was equipped with an integrated water heating system to maintain body temperature stability during the MR measurement. The head of the mouse was secured firmly by the front teeth into an anatomical nose–mouth mask, which prevented head movement and constantly supplied Isoflurane. Isoflurane levels were adjusted during the measurement (0.7–1.5%) to achieve a breathing rate of 90–120 breaths/min, which secured good BOLD-contrast and minimum head movement. The breathing rate was constantly monitored using a pediatric breathing sensor. The eyes were covered with an eye ointment (Bepanthen^®^, Bayer, Leverkusen, Germany) to prevent exsiccation damage. The dorsal side of both hind paws was fixed to a computer-controlled Peltier heating element each. A 3 cm 4-channel array head coil (Bruker Biospin MRI GmbH, Ettlingen, Germany) was fixed directly above the mouse brain for signal detection, ensuring good signal-to-noise-ratio.

All measurements were performed on our 4.7 T small animal Bruker Biospec 47/70 MRT (Bruker Biospin MRI GmbH, Ettlingen, Germany), equipped with a 200 mT/m gradient system and an actively-decoupled RF-coil-system for excitation. The scanner was operated using ParaVision V5.1 software from Bruker. After initial positioning of the cradle inside the scanner bore, stability of the mounting of the mouse brain was checked using a fast single shot gradient Echo Planar Imaging sequence (EPI, TR = 100 ms; TEeff = 25.3 ms; FOV = 15 mm × 15 mm; slice thickness 0.5 mm; matrix 64 × 64 voxel) of a single slice with 300 repetitions. Watched as a video, it allowed detecting of even small head movements. If the brain shifted more than one voxel, the animal was remounted. Next, a rapid acquisition relaxation enhanced sequence (RARE; TR = 2649 ms; TEeff = 56 ms; k-space averaging 4; RARE Factor = 8; FOV = 15 mm × 15 mm; 22 slices; slice thickness 0.5 mm; matrix 128 × 128 voxel, 16 coronal slices) was acquired and used as anatomical template for positioning of the axial volume (landmark: distal end of the lateral ventricle).

To ensure good image quality before the actual functional measurement, one volume of 22 axial slices covering the brain from Bregma -2.06 mm to 1.42 mm with an in-plane resolution of 0.234 mm × 0.234 mm was acquired using EPI (TR = 2000 ms; TEeff = 25.3 ms; FOV = 15 mm × 15 mm; slice thickness 0.5 mm; matrix 64 × 64 voxel; all further EPI measurements were run with these parameters). Only if quality was poor, local field inhomogeneities were adjusted using the FastmapScout macro implemented in ParaVision 5.1. Subsequently, a 10 min resting-state scan at identical slice positions was started.

After that, the thermal stimulation paradigm was run during the 74 min stimulus-driven EPI measurement (same settings but 2200 volumes): 3 sets of four ascending temperatures, innocuous 40 and 45 °C and noxious 50 and 55 °C, were presented alternatingly at both hind paws, starting with the right paw (Figure 1b). Temperatures were applied using Peltier heating elements controlled by an in-house developed software that allowed actively feed-back temperature management and exact coupling to the scanner trigger points for timely stimulation. The stimulus duration was 20 s (5 s ramp, 15 s plateau) and inter-stimulus interval was 3 min.

After the measurement, the mouse was removed from the scanner, the hind paws were treated with a cooling foam spray (Bepanthen^®^, Bayer, Leverkusen, Germany) and the mouse was placed on a tissue back in its home cage to wake up under visual control.

### 4.3. Data Analysis, Statistics and Blinding

#### 4.3.1. µCT and Biomechanical Tests

Graph Pad Prism 9.0 software was used for statistical analysis and graph preparation of results obtained from µCT analysis and biomechanical analysis. Data are represented as median ± 5/95 percentile. For µCT and biomechanical analysis, differences in median were analyzed using one-way ANOVA and Tukey’s post-hoc test. *p*-values ≤ 0.05 were considered as significant.

#### 4.3.2. Stimulus-Driven fMRI

Preprocessing of fMRI data was performed in BrainVoyager QX (V. 2.0.8; Brain Innovation BV, Maastricht, The Netherlands): after discarding the first two volumes due to saturation effects, preprocessing comprised motion correction to eliminate minimal brain movements (registration to first volume; trilinear detection and sinc interpolation), slice scan time correction (ascending interleaved, interpolation method cubic spline), spatial (Gaussian smoothing; kernel 2 pixel) and temporal smoothing (linear and non-linear high pass filtering, kernel 12s FWHM, FFT 9 cycles). Thereafter, voxel-wise GLM analysis (predictors folded by a mouse-specific two gamma HRF) was performed to assess the coupling strength between voxel time course and stimulation protocol. For each stimulus (i.e., paw–temperature combination, e.g., left paw 45 °C), one separate predictor was used to calculate stimulus-specific statistical parametric maps (SPMs) for each subject. Thus, eight maps were obtained per animal: one for each paw and stimulation temperature. All following analysis steps were performed in MagnAn (BioCom GbR, Uttenreuth, Germany).

For brain region-wise analysis, first, 22 slices that fitted best to our grey value fMRI slices were chosen from Paxinos mouse brain atlas [83]. From these slices, 211 brain regions (separate for left and right hemisphere) were selected, indexed with a unique identifier, color coded and converted into a digital 3D brain atlas. The atlas was adapted (translation x-y-z, scaling x-y, rotation z) to fit each single-measurement grey value anatomical fMRI separately, and converted for each measurement into structure-specific ROIs.

Single-subject SPMs were binarized and corrected for multiple comparisons using False Discovery Rate (FDR; q = 0.05, confined to brain mask). The remaining voxels were multiplied with the single-subject ROIs, resulting in labeled masks identifying the significantly activated voxels of 211 brain regions. Applying these labeled masks to the time series BOLD data, an average time profile of all significantly activated voxels per brain region were calculated. Conclusively, this step yielded average time profiles of all brain regions for each stimulus (predictor) per animal, separately. For the graph theoretical analysis, classically [84], a global signal regression (removal of the global mean) was performed subject-wise on the average time profiles. For each predictor and animal, a full adjacency matrix was obtained by calculating the Pearson correlation coefficient r between the average time profiles of all brain regions. Next, per predictor, a mean adjacency matrix across the animals was calculated after transforming the data into normally distributed Fisher-z-values and thereafter back into R-values. To ensure best topological comparison, the matrices were limited to the 5% strongest positive R-values.

#### 4.3.3. Resting-State fMRI

In accordance with the stimulus-driven fMRI data, the first two volumes were discarded. Resting-state preprocessing included only motion correction (registration to first brain volume; trilinear detection and sinc interpolation) and the slice scan time correction (ascending interleaved, interpolation method cubic spline). All further analyses, including FFT low pass filtering at 0.1 Hz, were performed using MagnAn (BioCom GbR, Uttenreuth, Germany).

Brain structure-specific ROIs were created for each subject as described above for the stimulus-driven fMRI data. Analysis was confined to voxels within the brain, by trimming the ROIs with individually drawn brain masks. For multi-seed region analysis (MSRA) [85], a seed region of 0.5 mm^2^ was automatically placed in the center of mass of each brain region. The Pearson correlation coefficient r was calculated between the average time course of each seed region and the single time courses of all voxels within the brain. Per animal, this produced one 3D correlation volume per seed region. Using FDR (q = 0.05), significant correlations were determined. After conversion to Fisher-z-values, the correlation values of all voxels that belonged to the same brain region were averaged for the given seed region, yielding one row in an asymmetric connectivity matrix per animal. Consequently, this was repeated for all brain regions. As described above, a mean adjacency matrix across the animals was calculated, and the data were back-transformed into R-values. Finally, the matrices were again limited to a density of the 5% strongest positive connections.

#### 4.3.4. Graph Theoretical Visualization

Connectivity matrices of the stimulus-driven and resting-state fMRI are displayed as 3D brain networks within an anatomical glass brain surface. The brain regions are represented by color-coded nodes at their anatomical position, and the correlation coefficients as edges between nodes. Size of the nodes represent the degree, i.e., how many connections (edges) this node has to all other nodes. Unconnected nodes are not displayed. Red or blue edges code the result of statistical tests; for exact, meaning see the legend of the corresponding figures. Visualization was carried out in Amira (V. 5.4.2 Thermo Fisher Scientific Inc., Waltham, MA, USA).

#### 4.3.5. Gene Expression Network Correlation Analysis

The Allen Brain Institute provides expression data for about 25,000 genes, aligned in a 200 µm voxel-grid spanning the whole mouse brain [26,86]. A corresponding 3D mouse brain atlas (Common Coordinate Framework, CCF version 3 [25]) allows to link expression data to the respective brain regions. In our case, the more than 800 partially very small brain structures of the CCF were merged into greater regions to match our Paxinos reference atlas used for fMRI analysis (see Section 4.3.2). Mean expression data per gene were calculated by averaging all grid voxels belonging to the corresponding brain region.

As the used mouse models differ in their expression of sensory and sympathetic neurotransmitters, we tried to attribute the differences found in the brain networks to certain neurotransmitters. Therefore, we used the gene expression data provided by the Allen Brain Institute. For the neurotransmitters α-CGRP, SP and NA, the brain region-specific mean expression data of the typical receptors (target) and proteins such as transporters and synthesis enzymes (origin) were calculated for each neurotransmitter. In case that more than one gene was related to the origin (or target) of the transmitter, the maximum expression per brain region of all related genes was used. For visualization purposes, the logarithm to the basis 2 of the maximal expression values was used. The FC networks were then split up to represent only brain regions as nodes that express either the origin or target genes and the corresponding edges. Hereby, we aimed to access information about which neurotransmitters influence the changes in FC most. Results of this network representation were summarized in tables for easier access (Table 1, Table 2 and Table 3).

#### 4.3.6. MRI Statistics

Significant differences in functional connectivity of stimulus-driven and resting-state data were assessed using homoscedastic or paired two-tailed Student’s *t*-test implemented in MagnAn, whichever was appropriate. The tests were permutation controlled (1000 permutations) for the family-wise error. Resulting *p*-values ≤ 0.05 were accepted as significant.

#### 4.3.7. Randomization and Blinding

The experimenter was not blinded due to differential treatment of the single animals regarding injections. Instead, all animals were analyzed at the same time, following a standardized procedure. The workflow does not require nor permit animal-specific input that may generate bias.

Special account was taken to use the different mouse models within one experimental group for MRI analysis at the same measurement day. Two experimental groups were operated (OVX/fracture) and measured (fMRI) at two consecutive days, respectively. The order, in which the mice of different experimental groups were measured over the course of one day, was randomized, to exclude daytime effects, but kept constant per single mouse over the three measurements.

## 5. Conclusions

The aim of this study was to investigate the influence of sensory and sympathetic neurotransmitters on fracture healing in a murine model of osteoporosis (OVX). To this end, we compared adult female mice lacking either sensory (Tac1−/− or α-CGRP−/−) or sympathetic (SYX) neurotransmitters with WT mice. Analysis included two parts: first, we assessed the callus microstructure and biomechanical properties of the fractured femur during the remodeling phase; second, we studied the development and resolution of fracture-associated hyperalgesia, focusing on the functional connectivity of the brain at rest and while processing noxious heat stimuli.

Micro-CT analysis revealed that the loss of sympathetic neurotransmitters had the most substantial effects on the microarchitecture of the callus in the remodeling phase. In contrast, loss of sensory neuropeptides had no major effect on callus maturation, but did lead to a general deterioration of bone structural properties. In both WT and SYX mice, incomplete bridging of the fracture site was observed in 40–50% of the animals.

We found that bone fracture modulated nociceptive brain functional connectivity specifically, enhancing amygdala–hippocampal connectivity shortly after fracture, which subsided mostly when the remodeling phase was reached. Resting state functional connectivity identified enhanced cortical and reduced cortical–medulla functional connectivity as unique hallmarks of fracture-induced hyperalgesia. Of all four groups, α-CGRP showed the most changes that lasted until the remodeling phase on day 21, indicating impaired resolution of hyperalgesia.

As a consequence, resting state functional connectivity between cortex and medulla may in the future prove to be a reliable biomarker for hyperalgesia and its temporal evolution induced by bone traumata such as fractures, improving therapy outcomes for the patient.

## Figures and Tables

**Figure 1 ijms-24-00510-f001:**
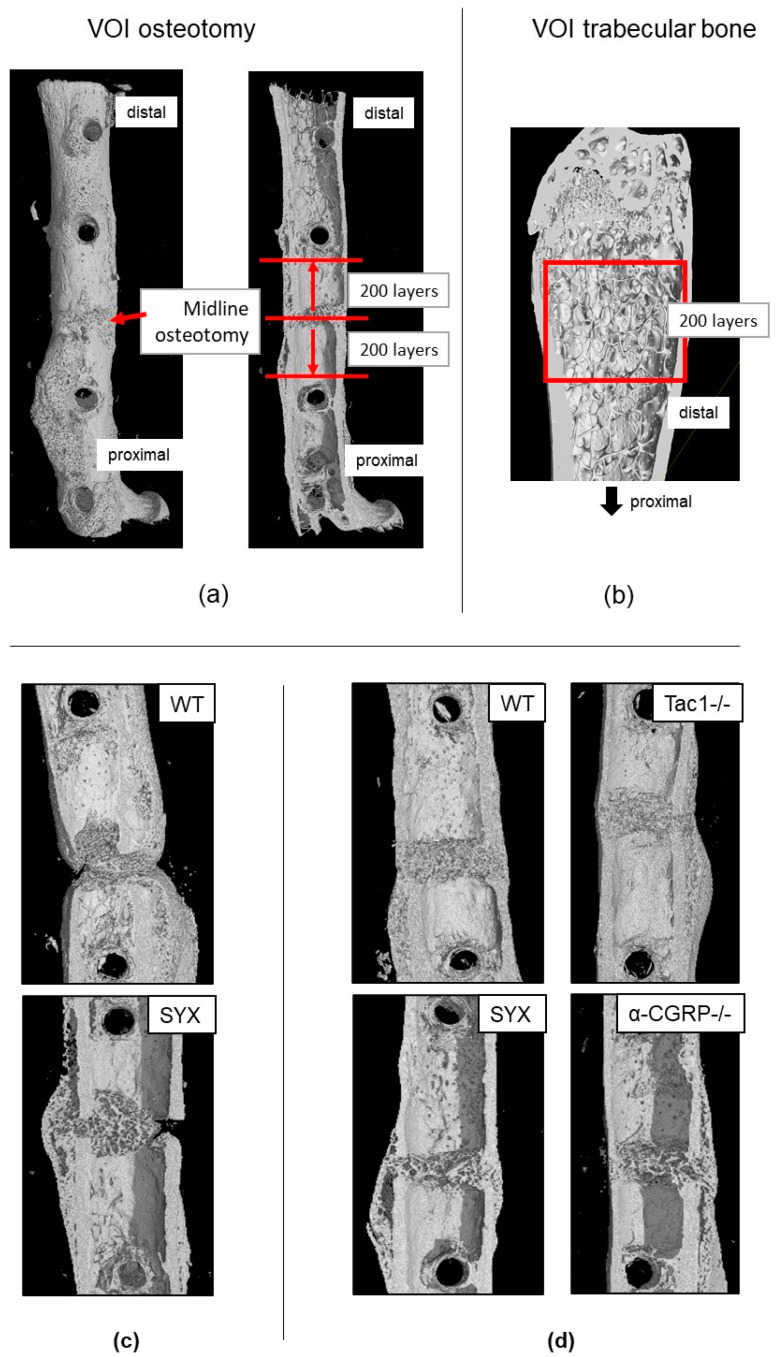
Representative images to demonstrate the region of interest (ROI) settings for µCT analysis and representative µCT scans of fractured femora from WT, SYX, Tac1−/− and α-CGRP−/− mice. (**a**) Volume of interest (VOI) of fractured femora 21 days after fracture. VOI was set 200 layers above and below the midline of the fracture, spanning in total 400 layers. (**b**) VOI of the trabecular bone in the distal part of the contralateral non-fractured femora started 50 layers from the cartilaginous bridge in the growth plate spanning a total of 200 layers. (**c**) Representative fracture site of the fractured femora of WT and SYX mice that collapsed (i.e., breakdown of diaphysis) during preparation for biomechanical analysis. (**d**) Representative fracture site of the fractured femora WT, SYX, Tac1−/− and α-CGRP−/− mice that underwent biomechanical analysis.

**Figure 2 ijms-24-00510-f002:**
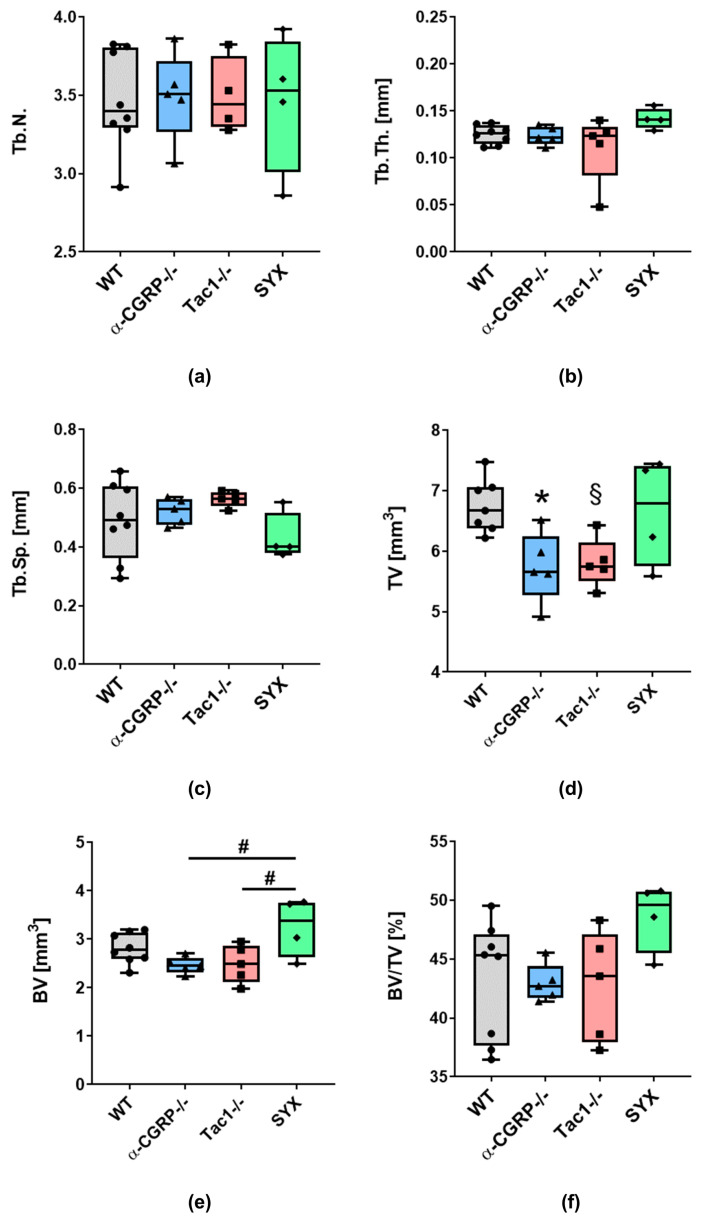
µCT analysis of fractured femora at 21 days after fracture of WT, α-CGRP−/−, Tac1−/−, and SYX mice. (**a**–**c**) Trabecular bone properties. Trabecular number (**a**), thickness (**b**) and separation (**c**) within the callus region of fractured femora. (**d**) Total volume (TV) of the callus region of fractured femora. (**e**) Bone volume (BV) of the newly formed trabecular bone structure and the cortical bone within the callus region of fractured femora. (**f**) Bone volume/total volume (BV/TV) within the callus region of fractured femora. * indicates differences between neurotransmitter-deficient groups and WT animals. # over lines indicate differences between neurotransmitter-deficient groups. */# = *p* ≤ 0.05. § indicates a trend compared to WT, § = *p* = 0.0532. (nWT = 8, nCGRP−/− = 5, nTac1−/− = 5, nSYX = 4).

**Figure 3 ijms-24-00510-f003:**
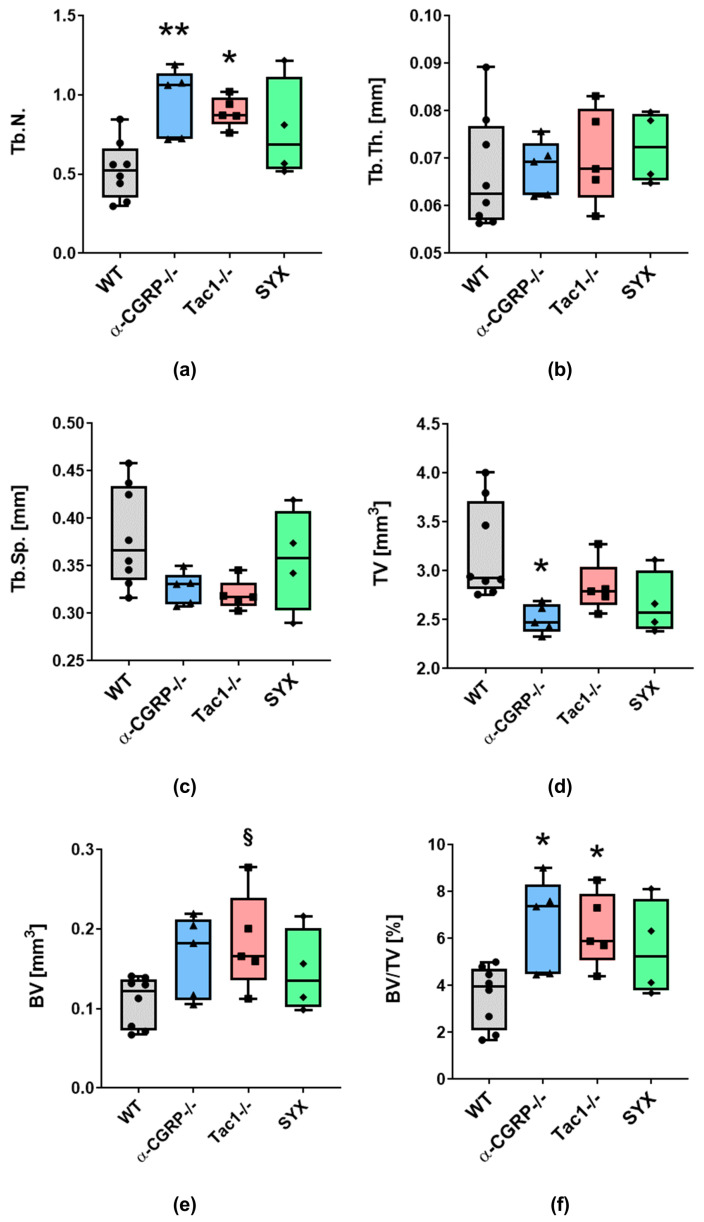
µCT analysis of the trabecular bone in the distal part of the contralateral non-fractured femora of WT, α-CGRP−/−, Tac1−/− and SYX mice at 21 days after fracture. (**a**–**c**) Trabecular bone properties. Trabecular number (**a**), thickness (**b**) and separation (**c**) of the trabecular bone in the distal region of the contralateral non-fractured femora. (**d**) Total volume (TV) of the trabecular bone in the distal region of the contralateral non-fractured femora. (**e**) Bone volume (BV) of the trabecular bone in the distal region of the contralateral non-fractured femora. (**f**) Bone volume/total volume (BV/TV) of the trabecular bone in the distal region of the contralateral non-fractured femora. * indicates differences between neurotransmitter-deficient groups and WT animals. * = *p* ≤ 0.05; ** = *p* < 0.01 § indicates a trend compared to WT, § = *p* = 0.0614. (nWT = 8, nCGRP−/− = 5, nTac1−/− = 5, nSYX = 4).

**Figure 4 ijms-24-00510-f004:**
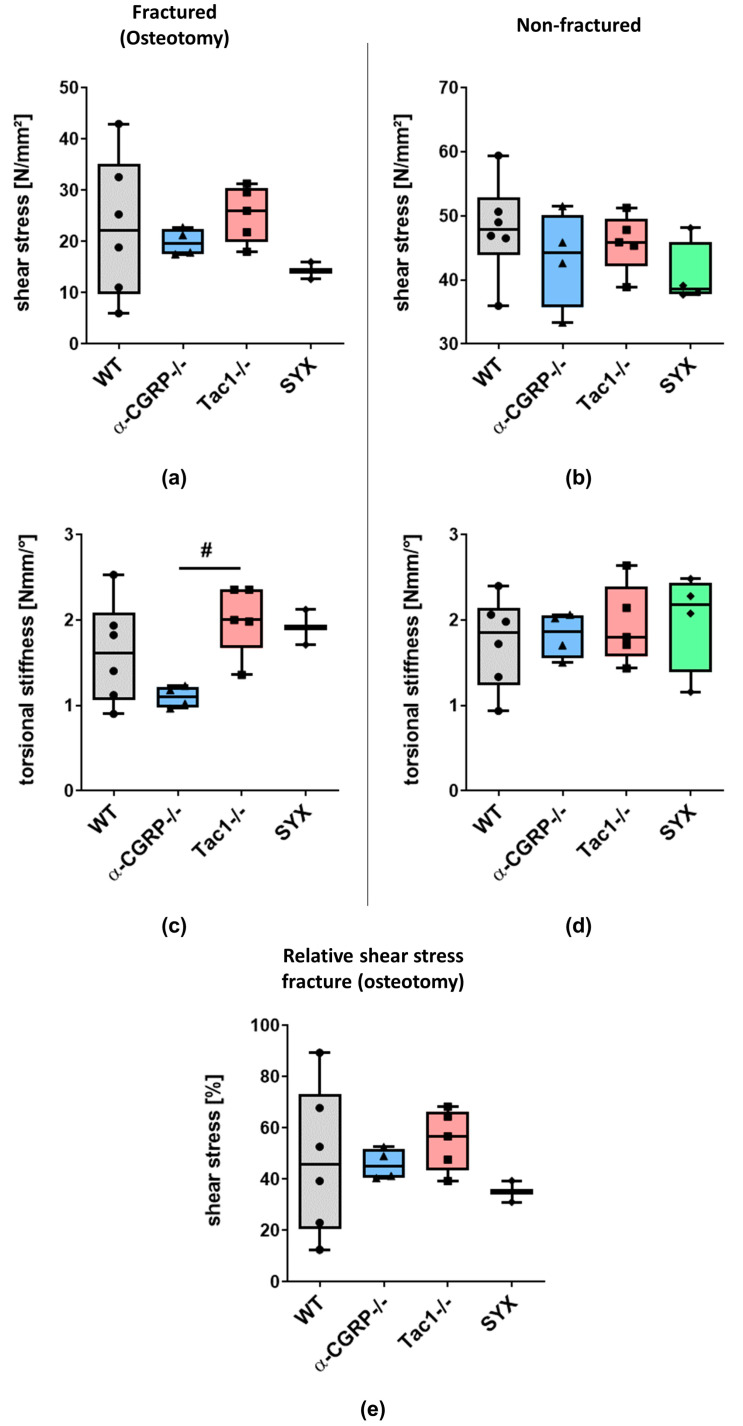
Biomechanical tests of fractured femora at 21 days after fracture and contralateral non-fractured femora from WT, α-CGRP−/−, Tac1−/− and SYX mice. (**a**,**b**) Shear stress (N/mm^2^) of fractured femora (**a**) and contralateral non-fractured femora (**b**). (**c**,**d**) Torsional stiffness (Nmm/°) of fractured femora (**c**) and contralateral non-fractured femora (**d**). (**e**) Relative shear stress of fractured femora, calculated as percentage of non-fractured femora (mean value of all non-fractured femora within one group). # over bars indicate differences between neurotransmitter-deficient groups. # = *p* ≤ 0.05. (nWT = 6, nCGRP−/− = 5, nTac1−/− = 5, nSYX = 2 fractured, 4 non-fractured).

**Figure 5 ijms-24-00510-f005:**
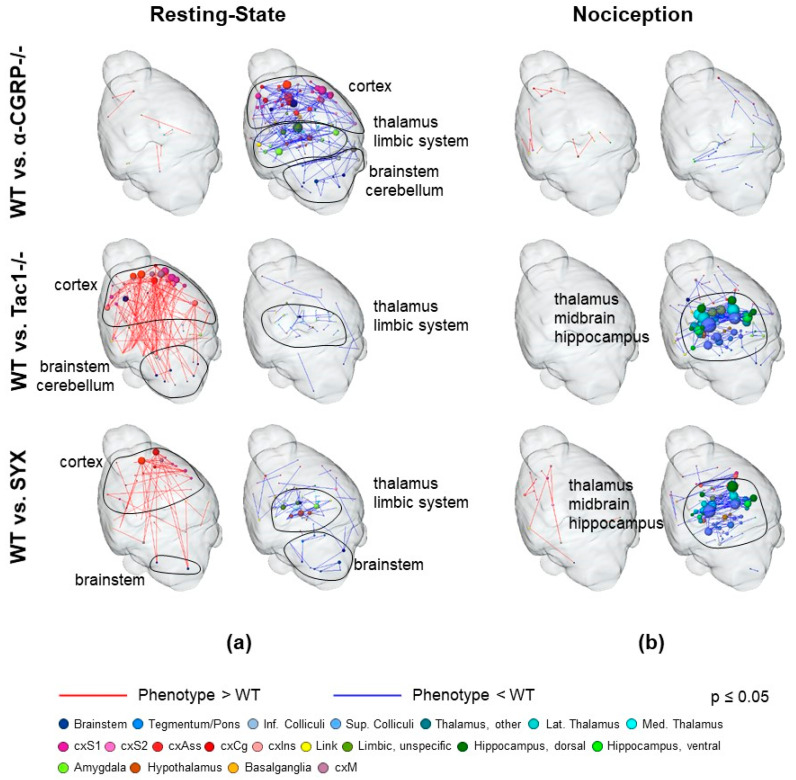
Baseline differences (day −2) in resting-state (RS) and nociceptive functional connectivity (FC) between the three neurotransmitter-deficient mouse models compared to WT. Significant changes in RS (**a**) and nociceptive (**b**) functional connectivity between the three neurotransmitter-deficient (NT-deficient) mouse models used in this study in comparison to WT mice are displayed by red (connectivity is increased) or blue (connectivity is decreased) edges between color-coded nodes as brain regions. The size of the node represents the degree, i.e., how many connections this node has to others. Unconnected nodes are omitted. Abbreviations: cxS1 (primary somatosensory cortex), cxS2 (secondary somatosensory cortex), cxAss (association cortex), cxCg (cingulate cortex), cxIns (insular cortex), cxM (motor cortex), inf. (inferior), lat. (lateral), med. (medial), sup. (superior). (nWT = 13, nCGRP−/− = 9, nTac1−/− = 9, nSYX = 8).

**Figure 6 ijms-24-00510-f006:**
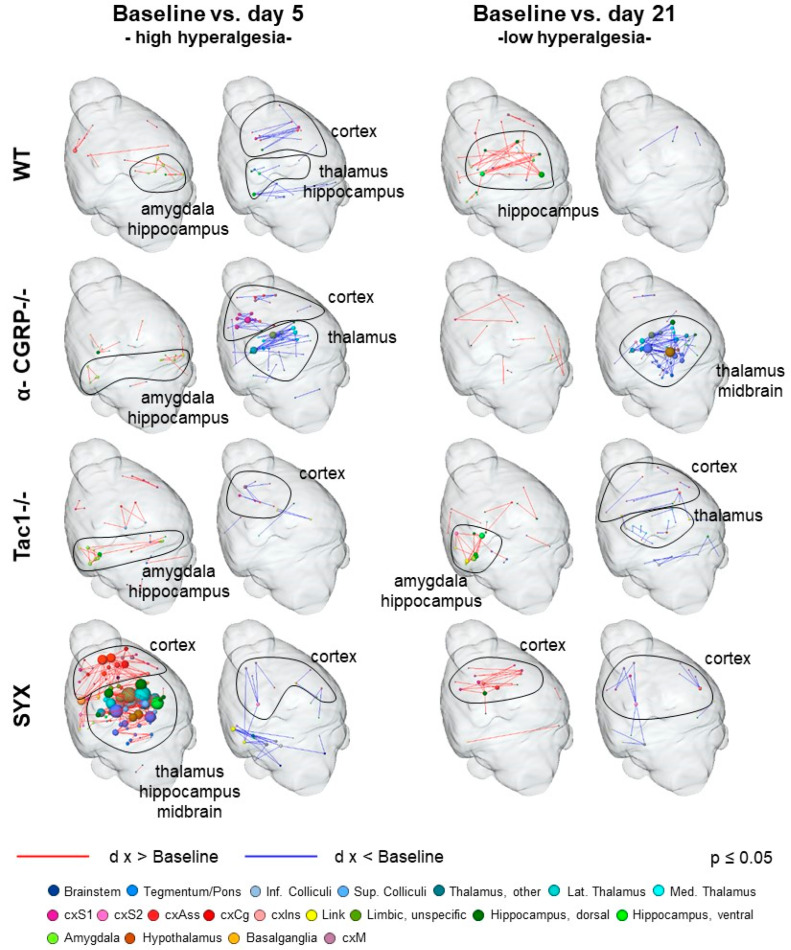
Enhanced amygdala and hippocampal connectivity reflected fracture-induced hyperalgesia evoked by nociceptive thermal stimulation of the hind paw of the fractured leg (noxious heat 50 and 55 °C, left paw). Red edges reflect enhanced and blue edges reflect decreased nociceptive FC at day 5 or day 21 after fracture, respectively, compared to baseline. Color-coded nodes represent the different brain regions, with the size of the nodes reflecting the degree, i.e., how many connections this node has to other unconnected nodes are omitted. Abbreviations: cxS1 (primary somatosensory cortex), cxS2 (secondary somatosensory cortex), cxAss (association cortex), cxCg (cingulate cortex), cxIns (insular cortex), cxM (motor cortex), inf. (inferior), lat. (lateral), med. (medial), sup. (superior). (nWT = 13, nCGRP−/− = 9, nTac1−/− = 9, nSYX = 8).

**Figure 7 ijms-24-00510-f007:**
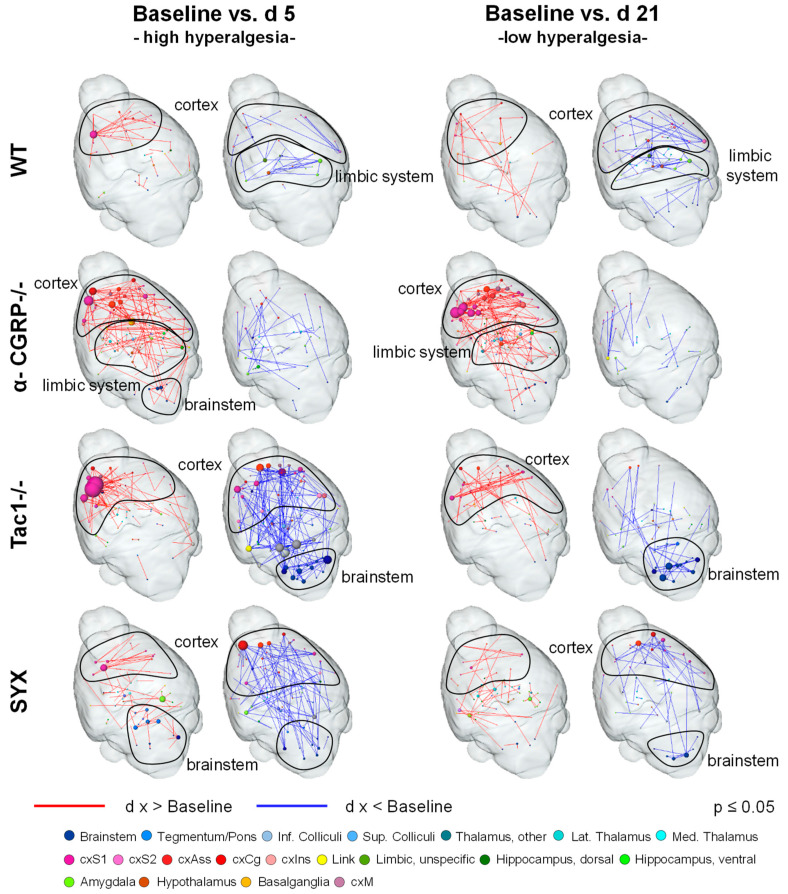
Functional connectivity of cortex and brainstem reflected cerebral hyperalgesia at rest as in vivo marker in resting-state brain networks. Red edges reflect enhanced, blue edges decreased resting-state functional connectivity (RS FC) at day 5 or day 21 after fracture, respectively, compared to baseline. Color-coded nodes represent the different brain regions, with the size of the nodes reflecting the degree, i.e., how many connections this node has to others unconnected nodes are omitted. Abbreviations: cxS1 (primary somatosensory cortex), cxS2 (secondary somatosensory cortex), cxAss (association cortex), cxCg (cingulate cortex), cxIns (insular cortex), cxM (motor cortex), inf. (inferior), lat. (lateral), med. (medial), sup. (superior). (nWT = 13, nCGRP−/− = 9, nTac1−/− = 9, nSYX = 8).

**Table 1 ijms-24-00510-t001:** Gene expression network correlation analysis data corresponding to Figure 5 (baseline differences in resting-state and nociceptive functional connectivity.

Resting State	Neurotransmitter	WT vs. α-CGRP−/−	WT vs. Tac1−/−	WT vs. SYX
	Noradrenaline	↓: Bs, cxAss, cxS	↑: cxAss, cxS, Hc	↑: cxAss, cxS ↓: Bs
	Substance P	↓: Bs, cxAss, Hy, Am, Bg	↑: cxAss ↓: Am	↑: cxAss, Am, Bg ↓: Bs, Hy, Am
	α-CGRP	↓: Hy, Bg		
Nociception	Neurotransmitter	WT vs. α-CGRP−/−	WT vs. Tac1−/−	WT vs. SYX
	Noradrenaline		↓: Th, Hc	↓: Th, Hc
	Substance P		↓: Bs, Hy, Am	↓: Bs, Hy, Am
	α-CGRP			

Gene expression network correlation analysis highlighted which neurotransmitters may be involved in the depicted differences. ↓: functional connectivity is reduced in the indicated mouse model compared to WT, and the brain regions express the indicated neurotransmitter. ↑: functional connectivity is enhanced in the indicated mouse model compared to WT, and the brain regions express the indicated neurotransmitter. Abbreviations: Am (amygdala), Bg (basalganglia), Bs (brainstem), cxS (somatosensory cortex), cxAss (association cortex), Hc (hippocampus), Hy (hypothalamus), Th (thalamus).

**Table 2 ijms-24-00510-t002:** Gene expression network correlation analysis data corresponding to Figure 6 (nociception). Data provide an overview of which neurotransmitters (of those modulated in our mouse models) are mainly expressed in the affected brain regions.

Baseline vs. D5	Neurotransmitter	WT	α-CGRP−/−	Tac1−/−	SYX
	Noradrenaline	↓: cxS, Hc	↑: Hc ↓: Th, cxAss, cxS	↑: cxS (right) ↓: cxS (left)	↑: Bs, Th, cxAss, cxS, Hc
	Substance P	↑: Am	↑: Am	↑: Bs, Am	↑: Bs, cxAss, Am, Bg
	α-CGRP				
Baseline vs. D21	Neurotransmitter	WT	α-CGRP−/−	Tac1−/−	SYX
	Noradrenaline	↑: cxS, Hc	↑: cxS ↓: Bs, Th, Hc	↑: Hc	↑: cxS, Hc
	Substance P	↑: cxAss, Hc, PAG	↓: Bs, PAG	↑: Bs, Am, Hc	↑: Hc
	α-CGRP				

↓: functional connectivity is reduced at day 5/21 compared to baseline, and the brain regions express the indicated neurotransmitter. ↑: functional connectivity is enhanced at day 5/21 compared to baseline, and the brain regions express the indicated neurotransmitter. Abbreviations: Am (amygdala), Bg (basalganglia), Bs (brainstem), cxS (somatosensory cortex), cxAss (association cortex), Hc (hippocampus), PAG (periaqueductal grey), Th (thalamus).

**Table 3 ijms-24-00510-t003:** Gene expression network correlation analysis data corresponding to Figure 7 (resting state). Data provide an overview of which neurotransmitters (of those modulated by our mouse models) are mainly expressed in the affected brain regions.

Baseline vs. D5	Neurotransmitter	WT	α-CGRP−/−	Tac1−/−	SYX
	Noradrenaline	↑: cxAss, cxS	↑: Bs, Th, cxAss, cxS, Hc	↑: cxAss (left) cxS (left) ↓: Bs, Th, cxAss (right), cxS (right), Hc	↑: Bs, Th, cxAss, cxS, Hc
	Substance P	↓: Am, Hy	↑: cxAss, Am, Hy, Bg ↓: Am	↓: cxAss, Bs, Am, Hy	↑: Bs, cxAss, Am, Hy, Bg
	α-CGRP			↓: Cb	↑: Cb
Baseline vs. D21	Neurotransmitter	WT	α-CGRP−/−	Tac1−/−	SYX
	Noradrenaline	↑: Bs, cxS	↑: Bs, Th, cxAss, cxS, Hc	↑: cxAss, cxS ↓: Bs, Hc	↑: Bs ↓: cxAss, cxS
	Substance P	↑: Bs, cxAss, Am, Bg ↓: Am	↑: Bs, cxAss, Am, Hy, Bg	↑: cxAss, Bg ↓: Bs, Am, Hy	↑: Am ↓: Bs, cxAss, Hy, Bg
	α-CGRP	↓: Cb	↑: Cb	↓: Cb	

↓: functional connectivity is reduced at days 5/21 compared to baseline, and the brain regions express the indicated neurotransmitter. ↑: functional connectivity is enhanced at day 5/21 compared to baseline, and the brain regions express the indicated neurotransmitter. Abbreviations: Am (amygdala), Bg (basalganglia), Bs (brainstem), Cb (cerebellum), cxS (somatosensory cortex), cxAss (association cortex), Hc (hippocampus), Hy (hypothalamus), Th (thalamus).

**Table 4 ijms-24-00510-t004:** Overview of the animal cohorts and numbers.

	Treatment Duration	−2 Days Before Fracture	5 Days Post Fracture	21 Days Post Fracture
Mouse Models				
Wildtype (WT)		13	13	5 (+3 of cohort 2 for biomechanical tests)
alpha-CGRP-*knockout* (α-CGRP−/−)		9	9	5
Tachykinin-1-*knockout* (Tac1−/−)		9	9	5
Sympathectomized WT (SYX)		8 (−1)	8 (−1)	4 (−1)

Numbers in columns represent the numbers of animals, used for fMRI analysis at time points −2 days before fracture, and 5 and 21 days post fracture. After time point day 5, animals were divided into two study cohorts—cohort 1 and cohort 2. Animal numbers of cohort 1 are listed at day 21 and were used for fMRI analysis plus µCT and biomechanical tests, addressed in the present study. Cohort 2 underwent a different treatment and will not be addressed here with one exception: femora of 3 WT animals were used for biomechanical tests on day 21 for the present study as replacement for femora of 2 WT mice that collapsed during preparation for biomechanical tests (i.e., breakdown of diaphysis at fracture site; see Group assignment and dropout; mice were numbered as +3 in brackets at day 21). One SYX mouse died during the first fMRI analysis at day −2 and had to be excluded from the study (numbered as −1 in brackets). Mouse model names in brackets refer to abbreviations used in the figures.

## Data Availability

µCT, biomechanical and fMRI raw data are available from the corresponding author upon reasonable request.

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
