# Peer review of "Influence of the Peripheral Nervous System on Murine Osteoporotic Fracture Healing and Fracture-Induced Hyperalgesia"

_ijms, 2022, doi:10.3390/ijms24010510_

Round 1

Reviewer 1 Report

The state-of-the-art in the field and need for this study are clearly described in the introduction. Conducted methods are described in detail and the results are clearly presented. I would emphasize the high quality of Figures. I would just recommend that authors add a scheme about the experimental design of the study to allow better understanding of the complex experimental design. 

The present paper is very interesting and will be of interest to the broad audience of International Journal of Molecular Sciences.

Author Response

We want to thank both reviewers for their positive and encouraging statements and critical suggestions in order to further improve this study!

Comment: I would emphasize the high quality of Figures. I would just recommend that authors add a scheme about the experimental design of the study to allow better understanding of the complex experimental design.

Response: We thank this reviewer again for the suggestions to improve the manuscript. All figures were additionally uploaded as .tiff files with a minimum of 300 dpi, so that the quality of the figures online will be according to the guidelines. We added a scheme about the experimental design (suppl. Figure S4), the reviewer made a good point here, this will help to improve the reader´s understanding of the timeline.

Reviewer 2 Report

The article entitled “Influence of the peripheral nervous system on murine osteoporotic fracture healing and fracture-induced hyperalgesia”. The aim was to characterize the development and resolution of fracture-associated hyperalgesia and fracture healing under impaired musculoskeletal conditions in adult female OVX mice.

Below are some suggestions:

In the Abstract:

- The abstract is clear and objective, but I suggest including the relevance and clinical importance of the research.

1. Introduction:

- The introduction is clear, but I suggest, if possible, better organize the ideas because they are often repetitive, making the reading more dynamic.

2. Results

- 2.1.1 µCT analysis:  I suggest, in general, separating the plates for better quality of and graphics

3. Discussion

- The discussion is well written, comparing the data found in the research with data from the literature. I suggest just entering the search limitations and if possible to summarize the discussion, it is an important part of the article and the reading is very tiring.

4. Materials and Methods

- The methodology is well described:

- 4.2.3. µCT protocols and analysis: Could the authors describe the reason for euthanasia at 21 days? Wouldn't that be too short a period?

5. Conclusions

- I suggest rewriting the conclusion because it is too long, bringing data from the results again. It can be started by making a general summary of what was done in the research with the objective, ending with the answer according to the presented objective, including the clinical applications.

Author Response

We want to thank both reviewers for their positive and encouraging statements and critical suggestions in order to further improve this study!

Comment: The abstract is clear and objective, but I suggest including the relevance and clinical importance of the research.

Response: Thank you for the suggestion. We added a comment about the clinical importance and relevance to the abstract section in lines 32-35.

Comment: -The introduction is clear, but I suggest, if possible, better organize the ideas because they are often repetitive, making the reading more dynamic.

Response: Thank you very much. We reordered parts of the introduction and deleted repetitive sentences to improve the reading of this section. Please see marked text in the introduction section.

Comment: 2.1.1 μCT analysis: I suggest, in general, separating the plates for better quality of and graphics

Response: We rearranged the graphs within the figures 2-4 (µCt and biomechanics) to improve the figure quality, the graphs are now bigger within each figure and the separate data points are better visible.

Comments: The discussion is well written, comparing the data found in the research with data from the literature. I suggest just entering the search limitations and if possible to summarize the discussion, it is an important part of the article and the reading is very tiring.

Response: Thank you for this valuable notion. We have already integrated limitations into the discussion (e.g. that two SYX and two WT femora broke during preparation for biomechanics, leading to replacement of WT femora but rendering SYX data unsuitable for statistical analysis; discussion of the influence of anesthesia on functional MRI data), and found that it would disrupt the reading flow too much if we moved the points to a separate section at the end.

We followed your suggestion and added an additional summary section to the end of the discussion to recapitulate the key findings.

Comment: The methodology is well described: 4.2.3. μCT protocols and analysis: Could the authors describe the reason for euthanasia at 21 days? Wouldn't that be too short a period?

Response: Thank you for asking. The present study was designed to analyze the effects of sensory and sympathetic neurotransmitter on fracture healing. Fracture healing can be separated into the induction phase (days 2-3 post murine fracture), the inflammatory phase (days 5-7 post fracture), the soft callus phase (days 8-12 post fracture), the ossification phase (days 13-16 post fracture) and the remodeling phase (days 17-21 post fracture) (see Figure 3 in Niedermair et al, 2020 (1)).  There might be some variations regarding the exact duration, nevertheless, nerve fibers start sprouting already into the fracture site within the first days during the inflammatory phase. Further, cells in the fracture callus are able to produce and react to neurotransmitter of the sympathetic and sensory nervous system (1,2). Effects of the sensory and sympathetic nerve fibers on fracture healing will therefore be visible when analyzing the remodeling phase at day 21 post fracture, when bone has been formed and is undergoing structural remodeling. Hence, we have chosen day 21 post fracture as an appropriate end point of the healing procedure. Hyperalgesia can be analyzed using fMRI at the time point of the inflammatory healing phase, where hyperalgesia might be high, and at the time point of remodeling, where hyperalgesia should be low under normal conditions. According to our experience, effects on callus maturation, delayed healing or non-union will be visible at day 21. Therefore, day 21 was considered the endpoint of our study, which also limits the duration of the experiment to what is necessary to address the aim of this study, with respect to the 3R principles (refinement).

Comment: I suggest rewriting the conclusion because it is too long, bringing data from the results again. It can be started by making a general summary of what was done in the research with the objective, ending with the answer according to the presented objective, including the clinical applications.

Response: Thank you for this recommendation. We rewrote the conclusion according to your suggestions, including the aim of the study, short overview over methods, summary of the results and implications for future clinical use. It is still quite long, but we find it a nice wrap-up of the study.

References

  1. Niedermair T, Straub RH, Brochhausen C, Grässel S. Impact of the Sensory and Sympathetic Nervous System on Fracture Healing in Ovariectomized Mice. International journal of molecular sciences. 2020;21(2). Epub 2020 Jan 8.
  2. Niedermair T, Kuhn V, Doranehgard F, et al. Absence of substance P and the sympathetic nervous system impact on bone structure and chondrocyte differentiation in an adult model of endochondral ossification. Matrix biology : journal of the International Society for Matrix Biology. 2014;38:22–35.